# Prevention of the Pro-Aggressive Effects of Ethanol-Intoxicated Mice by Schisandrin B

**DOI:** 10.3390/nu15081909

**Published:** 2023-04-15

**Authors:** Ho Yin Pekkle Lam, Ting-Ruei Liang, Shih-Yi Peng

**Affiliations:** 1Department of Biochemistry, School of Medicine, Tzu Chi University, Hualien 970, Taiwan; pekklelavabo@mail.tcu.edu.tw; 2PhD Program in Pharmacology and Toxicology, Tzu Chi University, Hualien 970, Taiwan

**Keywords:** alcoholic liver disease, liver fibrosis, hepatic encephalopathy, Schisandrin B

## Abstract

Excessive alcohol consumption can lead to serious health complications, with liver and neurological complications being the most important. In Western nations, alcoholic liver disease accounts for 50% of mortality from end-stage liver disease and is the second most common cause of liver transplants. In addition to direct damage, hepatic encephalopathy may also arise from alcohol consumption. However, effective treatment for liver disease, as well as neurological injury, is still lacking today; therefore, finding an efficacious alternative is urgently needed. In the current study, the preventive and therapeutic effects of Schisandrin B (Sch B) against ethanol-induced liver and brain injuries were investigated. By using two treatment models, our findings indicated that Sch B can effectively prevent and ameliorate alcoholic liver diseases, such as resolving liver injuries, lipid deposition, inflammasome activation, and fibrosis. Moreover, Sch B reverses brain damage and improves the neurological function of ethanol-treated mice. Therefore, Sch B may serve as a potential treatment option for liver diseases, as well as subsequential brain injuries. Furthermore, Sch B may be useful in preventive drug therapy against alcohol-related diseases.

## 1. Introduction

Excessive and/or chronic alcohol consumption can lead to serious health complications, affecting almost every organ in the body. Alcoholism is highly related to gastrointestinal bleeding [1,2], pancreatitis [3], alcoholic liver disease [4,5], neurologic disorders [6], type II diabetes mellitus [7], and cancers [8,9,10]. In Western nations, alcoholic liver disease accounts for 50% of mortality from end-stage liver disease and is the second most common cause of liver transplants [11]. While the liver is the major organ responsible for the processing of alcohol [12], excessive alcohol may destroy liver cells, leading to inflammation and ultimately fibrosis. While early and slight liver problems may be treated with lifestyle modifications, effective treatment for chronic and late-stage liver problems is still lacking today [13]. In conjunction with the direct damage induced by alcohol, the brain can be affected by the damaged liver [14]. Hepatic encephalopathy is a central nervous system disorder that occurs when the liver does not function properly and, as a result, unprocessed toxins build up and travel to the brain [15]. Hepatic encephalopathy is usually associated with poor quality of life and a high mortality rate [15], so finding an effective therapy for alcohol-induced liver disease, as well as its associated encephalopathy, is urgently needed.

It has been reported that Schisandrin B (Sch B), an extract of *Schisandra chinensis*, protects against different causes of liver injuries [16,17,18,19,20]. Sch B has also been shown to improve damage to different organs, mainly through its anti-inflammatory [16,19,21], anti-oxidative [16,22], and anti-fibrotic [17,18,19] properties. Currently, Sch B has been shown to protect against acute alcoholic cardiomyopathy through downregulation of autophagy [23]. Sch B also enhances antioxidant status in the mitochondria of multiple tissues in mice with chronic ethanol feeding [24]. However, there seems to be a lack of focus on Sch B in ethanol-induced liver injuries.

Moreover, despite many studies focused on therapeutics for alcohol-induced liver injuries, only a few studies have examined brain damage caused by alcohol consumption. In this study, we examined the beneficial effects of Sch B on alcohol-induced liver and brain damage.

## 2. Materials and Methods

### 2.1. Drugs

Schisandrin B (Sch B; Chengdu Alfa Biotechnology, Chengdu, China; Figure 1) was dissolved in olive oil to achieve a concentration of 20 mg/kg. Ethanol was diluted with distilled water to yield the following concentrations: 30% (*v/v*), 40% (*v/v*), and 50% (*v/v*).

### 2.2. Animals

Eight-week-old male BALB/c mice were obtained from the National Laboratory Animal Center, Taipei, Taiwan. Mice were housed in the animal center at Tzu Chi University and maintained in 23 °C ± 1 °C with a 12 h light/dark cycle and 40–60% humidity conditions, along with an ad libitum diet. All experimental procedures involving animals were approved by the Institutional Animal Care and Use Committee (IACUC) of Tzu Chi University (No. 109066).

### 2.3. Animal Treatment

Two treatment models (preventive model and corrective model) were employed in this study to investigate the effect of Sch B. In the first model (preventive model), 16 mice were randomly divided into 4 groups—control, Sch B-treated control, ethanol-treated, and ethanol + Sch B-treated—with each group containing 4 mice. Ethanol was orally administered to the mice as follows: 35% ethanol for 7 days, 40% ethanol for the following 7 days, and a 52% ethanol binge on the 15th day. Sch B (20 mg/kg) was orally administered 1 h after ethanol administration for the same duration. Mice were subjected to neurofunctional tests before the administration of a new ethanol dose. Mice were sacrificed one day after the last treatment with 12 h fasting. In the second treatment model (corrective model), 16 mice were randomly divided into 4 groups—control, Sch B-treated control, ethanol-treated, and ethanol + Sch B-treated—with each group containing 4 mice. Mice were first administered ethanol in the way mentioned above. One day after the 52% ethanol binge, mice were treated with 20 mg/kg Sch B by oral gavage for 14 continuous days. Mice were sacrificed by exsanguination without anesthesia one day after the last Sch B treatment (day 30) with 12 h fasting. The method for ethanol treatment was employed as described previously [25,26,27]. Treatment concentrations of Sch B (20 mg/kg) were based on literature references [28,29]. No mice died or exhibited Sch B-related adverse effects during the experiment.

### 2.4. Hematoxylin & Eosin (H&E) and Sirius Red Staining

Livers and brains were fixed in 10% neutral buffered formalin and dehydrated in a series of alcohols. After immersion, tissue blocks were cut into sections approximately 5–8 micrometers thick and stained with hematoxylin and eosin (H&E) or picro-Sirius red, as described previously [19].

### 2.5. Histopathological Examination and Scoring

Histopathological liver changes were evaluated as follows: 0, no injuries; 1, mild injuries with cytoplasmic vacuolization and slight pyknosis; 2, moderate injuries with extensive pyknosis and loss of cellular borders; and 3, severe injuries with disintegration of hepatocytes, tissue congestion, and cellular infiltration. Brain sections were scored as follows: 0, no evidence of injuries; 1, dispersed pyknosis; 2, slight necrosis or apoptosis; and 3, multiple or extensive necrosis or apoptosis [30]. At least 10 random fields were scored in each liver section, and five random fields in each brain section were scored. The severity of liver fibrosis was also evaluated by Ishak scoring of the picro-Sirius red-stained slides [31].

### 2.6. Western Blotting

Total protein was extracted from tissues, resolved by sodium dodecyl sulfate (SDS)-polyacrylamide gel electrophoresis (PAGE), and transferred to a polyvinylidene difluoride (PVDF) membrane (EMD Millipore, Danvers, MA, USA). After blocking, the membranes were incubated overnight with antibodies against α-tubulin (GeneTex, Irvine, CA, USA), type I collagen (COLA-1; ABclonal, Woburn, MA, USA), type III collagen (COLA-3; ABclonal), fibronectin (GeneTex), NOD-, LRR-, and pyrin domain-containing protein 3 (NLRP3; Proteintech, Rosemont, IL, USA), caspase-1 (Proteintech), interleukin-1β (IL-1β; Cell Signaling Technology, Danvers, MA, USA), interleukin-18 (IL-18; Proteintech), and gasdermin D (GSDMD; Santa Cruz Biotechnology, Dallas, TX, USA). The membranes were then incubated with horseradish peroxidase (HRP)-conjugated secondary antibodies (EMD Millipore) for 1 h. Protein bands were visualized with enhanced chemiluminescence (ECL) reagents (EMD Millipore) and quantified using densitometry with Image J software (v1.46). The band intensities of the proteins of interest were normalized by that of α-tubulin or the uncleaved band.

### 2.7. Collection of Serum, Cerebrospinal Fluid (CSF), and Liver Lysate

A cardiac puncture was performed to obtain blood samples from the mice. Samples were allowed to clot for 30 min, followed by centrifugation for 15 min at 600× *g* to collect serum. Cerebrospinal fluid (CSF) was collected by injecting ice-cold sterile phosphate-buffered saline (PBS) into the cranial cavity and cerebral ventricles. The washing solution was collected and centrifuged at 600× *g* for 10 min and then later at 10,000× *g* for 30 min. The obtained supernatant was stored at −80 °C until use. Liver lysate was obtained by homogenizing a weighted fraction of the liver at 4 °C in sterile PBS using a tissue homogenizer. Homogenates were centrifuged at 1500× *g* at 4 °C for 15 min, and supernatants were stored at −80 °C.

### 2.8. Measurement of Biochemical Parameters

Alanine transaminase (ALT), aspartate aminotransferase (AST), total cholesterol, triacylglycerides, and albumin were analyzed in serum and CSF using a Hitachi 7080 chemistry analyzer (Hitachi Ltd., Tokyo, Japan). Liver lysate was analyzed for total cholesterol and triacylglycerides using a cholesterol liquicolor kit (HUMAN Diagnostics Worldwide, Wiesbaden, Germany) and a triglycerides liquicolor kit (HUMAN Diagnostics Worldwide). Levels of transforming growth factor-beta (TGF-β) in the serum and liver lysate were measured using a mouse TGF-β sandwich enzyme-linked immunosorbent assay (ELISA) kit (Thermo Fisher Scientific, Waltham, MA, USA).

### 2.9. Neurofunctional Tests

Mice were subjected to neurofunctional tests before the beginning of the experiment, before administration of a new ethanol dose, and every 7 days during Sch B treatment. The wire hang test was performed by hanging a mouse upside down on a wire mesh. The time when the mouse fell was documented within 120 s. The beam walk test was performed by placing a mouse on a beam of 1 cm × 70 cm and 50 cm high above the platform. The time for the mouse to walk across the beam was recorded within 120 s. Clasping scores were allocated by hanging a mouse’s tail for 10 s. A score was recorded as follows: 0, the mouse showed normal escape extension; 1, the hindlimb of the mouse withdrew but did not touch the abdomen; 2, the hindlimb of the mouse withdrew and touched the abdomen but was not clasped; 3, the mouse showed immobility, with hindlimbs clasped and touching the abdomen. The hot plate test was performed by placing a mouse on a hot plate at 55 °C, and the time for their first paw reaction was documented within 15 s. The vertical pole test was performed by placing a mouse at the top of a 50 cm vertical pole. The time for the mouse to turn around and climb down was documented within 120 s. Each mouse was tested three times in each experiment.

### 2.10. Statistical Analysis

All results in this study were analyzed using GraphPad Prism 6.01 software (GraphPad Software, San Diego, CA, USA) and shown as mean ± standard deviation (SD), unless stated otherwise. Statistical analysis was conducted using one-way analysis of variance (ANOVA), and pairwise differences were evaluated using a post hoc test.

## 3. Results

### 3.1. Schisandrin B Prevents Liver Injuries in Mice Consuming Ethanol

To explore the protective effect of Sch B against ethanol intoxication, mice were treated with ethanol, as well as Sch B, on the same day (Figure 2A). Histopathological examination showed that ethanol consumption significantly altered hepatic cord arrangements, which was accompanied by congestion and ballooning degeneration (Figure 2B,C). Additionally, ethanol consumption significantly induced serum ALT and AST levels (Figure 2D,E), suggesting liver injury. Excessive ethanol consumption has been shown to alter lipid metabolism in the liver, leading to the accumulation of hepatic lipids and steatosis [5]; total cholesterol and triacylglyceride levels were measured in both the serum and liver. The results revealed that ethanol consumption, although it did not significantly increase total cholesterol and triacylglyceride levels in the serum, significantly increased their levels in the liver (Figure 2F–I). Sch B treatment, on the other hand, showed significant protection against ethanol-induced liver injuries, as shown by improvements in liver histology (Figure 2B,C), as well as ALT and AST levels (Figure 2D,E). Total cholesterol and triacylglyceride levels were also decreased in both serum and liver compared to ethanol-treated mice (Figure 2F–I). In the experiment, Sch B treatment alone had no adverse effect on the mice’s livers. Therefore, the use of Sch B can prevent liver damage as well as lipid accumulation caused by ethanol consumption.

### 3.2. Schisandrin B Prevents Activation of Liver Inflammasomes in Mice Consuming Ethanol

Ethanol intake has been linked to liver fibrosis [4,5]; therefore, Sirius red staining was performed to analyze collagen deposition in the liver. The results suggest that ethanol consumption, although it did not cause fibrosis, significantly increased collagen deposits in the liver (Figure 3A). Levels of TGF-β, a powerful profibrotic cytokine, were measured in the serum and liver of ethanol-fed mice, with levels found to be significantly increased in the liver after ethanol consumption (Appendix A). Western blotting also confirmed that ethanol increases type III collagen expression as well as fibronectin expression in the liver (Figure 3B). Inflammasome-induced inflammation has been shown to be positively correlated with fibrosis [32], so markers of inflammasomes were analyzed. Inflammasome components, including NLRP3 and caspase-1, as well as their effectors, interleukin (IL)-1β and IL-18, significantly increased due to ethanol consumption compared to the control and Sch B groups (Figure 3C). GSDMD, a pyroptotic marker that can be initiated by NLRP3 signaling [33], was also significantly increased in expression by ethanol, suggesting the presence of pyroptosis (Figure 3C).

Sch B treatment, when used during chronic ethanol consumption, did not prevent collagen deposition in the liver (Figure 3A). ELISA and Western blotting also showed similar results, with only fibronectin expression slightly decreased compared to the ethanol-treated group (Figure 3B and Appendix A). In the experiment, Sch B treatment blocked ethanol-induced inflammasome activation (Figure 3C). However, Sch B failed to inhibit pyroptosis in this treatment model, as suggested by the unchanged level of GSDMD. These results showed that Sch B, when used along with alcohol consumption, can prevent ethanol-induced inflammasome activation, but not fibrosis.

### 3.3. Schisandrin B Prevents Neurological Function Defects in Mice Consuming Ethanol

Ethanol consumption has always been shown to cause neurological defects [6]. To explore whether Sch B can prevent ethanol-caused neurological impairments, mice were subjected to different neurofunctional tests. As indicated, ethanol intake significantly decreased the mice’s weight (Figure 4A) and altered the mice’s neurological performances (Figure 4B–F). Sch B treatment was effective at improving some aspects of neurological performance, such as the hot plate test, ledge beam test, wire hang test, and vertical pole test. However, Sch B treatment alone did not alter the neurological functions of the mice compared to the control group.

### 3.4. Schisandrin B Prevents Brain Injuries in Mice Consuming Ethanol

As Sch B treatment improved neurological performance, we therefore evaluated the degree of brain injuries in ethanol-intoxicated mice, as well as the protection offered by Sch B. Histological analysis revealed numerous apoptotic neurons and hemorrhages in the brains of ethanol-drinking mice (Figure 5A,B). Ethanol consumption also increased CSF LDH levels (Figure 5C). Albumin synthesis occurs in the liver but not in the CNS, so albumin present in CSF is usually derived from plasma [28,34]. The ratio of CSF to serum albumin was used to evaluate blood-brain barrier (BBB) damage. The results showed that ethanol intake caused a significant increase in CSF albumin and CSF to serum albumin ratio, suggesting BBB damage (Figure 5D,F). To be noted, the decrease in serum albumin in ethanol-treated mice also suggests liver injury, as albumin synthesis mainly takes place in the liver (Figure 5D). Although Sch B treatment reduced the number of apoptotic neurons (Figure 5A,B), it did not affect the level of LDH (Figure 5C), which may suggest the occurrence of persistent injuries. On the contrary, Sch B decreased the level of CSF albumin, as well as the albumin ratio (Figure 5D,F), suggesting that the blood-brain barrier (BBB) was spared from damage. These results therefore suggest that Sch B, in addition to ethanol-induced liver injuries, protects against ethanol-induced neuropathy.

### 3.5. Schisandrin B Alleviates Ethanol-Induced Liver Injuries in Mice

Next, we wanted to determine whether Sch B can reverse already-caused ethanol-induced liver injuries. Therefore, mice were first treated with ethanol to induce injuries in the liver and brain and then with Sch B following chronological sequence (Figure 6A). Similar to what was observed, ethanol treatment caused significant liver injuries, including congestion and ballooning degeneration, which was reversed by Sch B treatment (Figure 6B,C). This improvement was accompanied by decreased serum ALT and AST levels (Figure 6 D,E) compared with ethanol-treated mice. Sch B treatment also beneficially lowered total cholesterol and triacylglycerides in both serum and liver. Furthermore, by comparing the ethanol-fed group with the treatment group, we can observe that ethanol-induced injuries persisted, suggesting that there was no natural reversion once the ethanol was removed. These results therefore suggest that Sch B can alleviate ethanol-induced liver injuries.

### 3.6. Schisandrin B Alleviates Ethanol-Induced Liver Fibrosis, Inflammasome Activation, and Pyroptosis

Additionally, Sch B treatment significantly reduced collagen deposition (Figure 7A) along with TGF-β (Appendix A), collagen III, and fibronectin (Figure 7B) expression in the liver of ethanol-treated mice. Ethanol-induced inflammasome activation, as well as pyroptosis, was also suppressed by Sch B treatment (Figure 7C). Therefore, Sch B may be promising in preventing ethanol-induced liver inflammation and fibrosis.

### 3.7. Schisandrin B Repairs Ethanol-Induced Neurofunctional Defects by Alleviation of Brain Injuries in Mice

Sch B treatment slightly but not statistically significantly increased the mice’s weight compared to ethanol-treated mice (Figure 8A). Except for the hot plate test, which showed only limited beneficial improvements, Sch B-treated mice showed beneficial improvements in the other neurological tests (Figure 8B–F), suggesting that Sch B can partially repair ethanol-induced neurofunctional defects in mice.

Finally, Sch B treatment significantly reduced ethanol-induced neuron cell death (Figure 9A,B) and improved reduced CSF LDH levels (Figure 9C). BBB damage was also significantly repaired by Sch B, as suggested by the reduced CSF-to-serum albumin ratio compared with the ethanol-treated group (Figure 9D–F).

## 4. Discussion

Chronic and excessive ethanol consumption may lead to intoxication and the development of chronic diseases, with liver disease being the most common and predominant [4,5]. Furthermore, excessive ethanol consumption can lead to structural and functional abnormalities in the brain [6]. In this study, we employed two treatment models to show that Sch B can be effective in preventing and treating ethanol-induced liver and brain injuries.

In the preventive treatment model, we explored the preventive effect of Sch B on ethanol-induced injuries by feeding the mice with both ethanol and Sch B on the same day (Figure 2A). The results suggested that Sch B was able to prevent liver damage (Figure 2B–E) as well as lipid deposition (Figure 2F–I). Inflammasomes are one of the key factors in liver fibrogenesis [32]. In the wake of sensing danger signals, NLRP3 activates caspase-1, thereby activating and releasing the inflammatory cytokines IL-1β and IL-18 [35]. In addition, it initiates a form of programmed cell death called pyroptosis, which is driven by a protein called GSDMD [36]. Once the cell dies, it activates Kupffer cells, which then release TGF-β and activate hepatic stellate cells (HSCs) [32,37]. The activated HSCs activate the downstream TGF-β/SMAD signaling pathway and cause fibrosis [38]. In addition to TGF-β, IL-1β also plays a role in a similar way [32]. While Sch B has some effect on inflammasome activation (Figure 2C), it cannot prevent collagen deposition in the liver (Figure 3A,B).

Since ethanol consumption also induces neurological defects [6,39], we next focused on the mice’s neurological behavior as well as brain pathology. Several neurological tests were employed in this study: the hot plate test was used to measure thermal nociception in the mice, the ledged beam test was used to assess sensorimotor deficits in the mice, the hindlimb clasping score was used as an indicator of the severity of motor dysfunction, the wire hang test was used to evaluate motor functions in the mice, and the pole test was employed to assess basal ganglia-related movement in the mice [40]. Our results suggest that Sch B treatment can beneficially prevent ethanol-induced neurological defects (Figure 4). In contrast to our study, in a mouse model with *Angiostrongylus cantonensis*-induced meningoencephalitis, Sch B by itself failed to repair the mice’s neurological functions; it was only effective if used together with the standard parasitic treatment drug albendazole [40]. This could be because using Sch B alone could not kill the parasites; therefore, even if Sch B ameliorated brain inflammation, neurological functions did not improve, as the injuries were persistent [40]. Another study using a mouse model with *Schistosoma manosni* (*S. mansoni*)-associated neurological deflects showed another therapeutic outcome from Sch B [41]. *S. manosni* is a parasitic worm that causes liver fibrosis, and liver fibrosis always leads to subsequent neurological damage [42]. In that study, the use of Sch B alone could already improve mice’s performance on neurological tests [41]. As Sch B resolved liver injuries and fibrosis in *S. mansoni*-infected mice, it led to less severe brain injuries, thereby improving the mice’s neurological functions. Along with neurological functions, improvements in brain pathology (Figure 5A–C) and BBB damage (Figure 5D–F) were also observed. Previously, the beneficial effects of Sch B have been suggested in several CNS diseases, such as cerebral ischemia [43,44], cerebral artery occlusion [44], subarachnoid hemorrhage [45], chemical-induced brain injury [28], and parasite-induced brain injury [40]. These studies have suggested that Sch B may target inflammasome activation to ameliorate brain inflammation and injuries. Therefore, it is presumed that Sch B may not only hinder inflammasome activation to prevent ethanol-induced neurological injuries, but this preventive effect may also come as a result of the resolution of liver damage.

Our second model was employed to investigate whether Sch B can be used to treat ethanol-induced injuries that have already been caused (corrective model). Surprisingly, Sch B worked better in treating than preventing ethanol-induced liver injuries (Figure 6). This treatment effect on the liver has also been observed in chemical-induced [28] and infection-related liver injuries [19]. Sch B treatment also successfully inhibited inflammasome activation as well as collagen deposition (Figure 7). Straightforwardly, unlike the first model, there was no consistent ethanol damage; therefore, Sch B can successfully perform its therapeutic functions. Sch B also improved neurological functions in ethanol-intoxicated mice (Figure 8), which was associated with improvements in neurological pathology and less BBB disruption (Figure 9).

In conclusion, our results showed that Sch B ameliorates liver injuries, lipid deposition, inflammasome activation, and fibrosis. Moreover, Sch B reverses brain damage and improves the neurological function of ethanol-treated mice. While effective treatment for liver injury is still lacking today, Sch B may serve as a potential treatment option for liver diseases as well as subsequential brain injuries. Furthermore, Sch B may be useful in preventive drug therapy against liver injuries.

## Figures and Tables

**Figure 1 nutrients-15-01909-f001:**
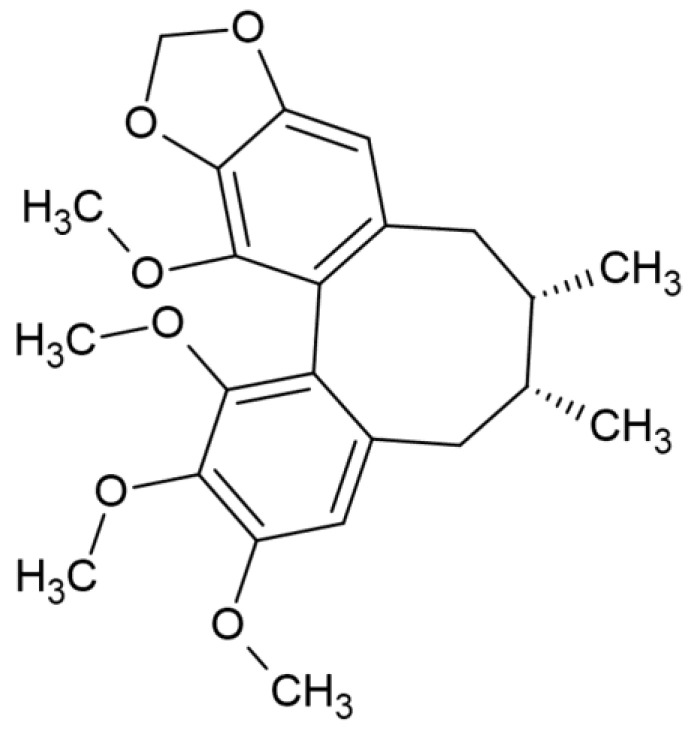
The chemical structure of Schisandrin B.

**Figure 2 nutrients-15-01909-f002:**
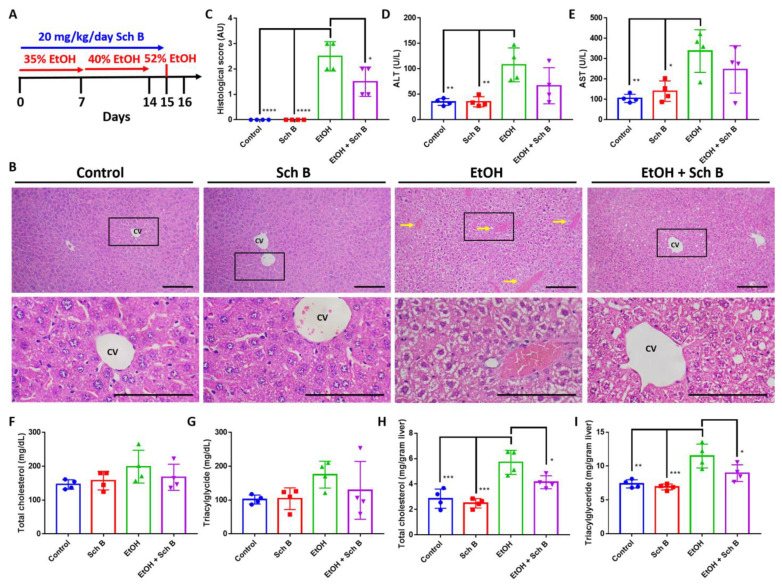
Schisandrin B prevents liver injuries in mice consuming ethanol. (**A**) Experimental design scheme. (**B**) Representative H&E-stained histological images showing the liver pathology of the control, Sch B-treated, ethanol-challenged, and ethanol-challenged + Sch B-treated mice. Congestion (yellow arrows) and ballooning degeneration were seen in the ethanol-treated mice, which improved after Sch B treatment. The scale bar corresponds to 200 μm. CV, central vein. (**C**) Histological scores based on the histological slides from (**B**). Serum levels of (**D**) ALT, (**E**) AST, (**F**) total cholesterol, and (**G**) triacylglycerides. Hepatic levels of (**H**) total cholesterol and (**I**) triacylglycerides. *n* = 4 mice. Results are expressed as mean ± SD. * *p <* 0.05, ** *p <* 0.01, *** *p <* 0.001, and **** *p <* 0.0001.

**Figure 3 nutrients-15-01909-f003:**
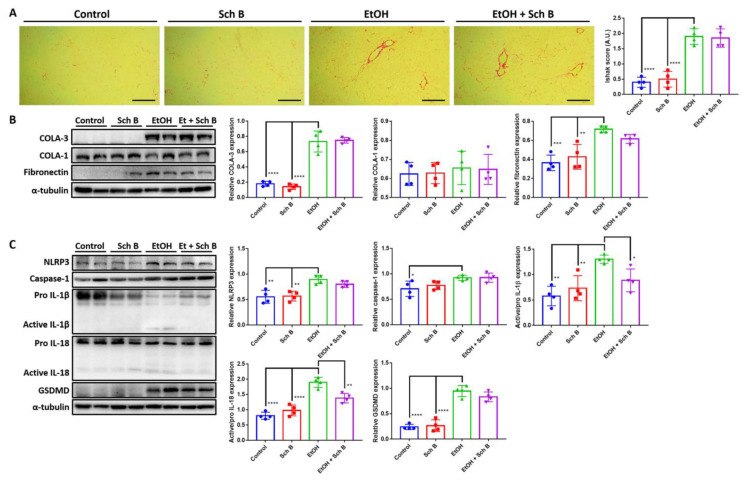
Schisandrin B prevents activation of liver inflammasomes in mice consuming ethanol. (**A**) Representative Sirius red-stained histological images showing collagen deposition (red color) in the livers of control, Sch B-treated, ethanol-challenged, and ethanol-challenged + Sch B-treated mice. Scale bar corresponds to 200 μm. Levels of fibrosis were quantified by Ishak score based on the Sirius red-stained slides. (**B**) Representative Western blot images and relative densitometric bar graphs of (**B**) fibrotic markers and (**C**) inflammasome components. Expression levels are standardized by α-tubulin or the pro-form protein levels as appropriate. *n* = 4 mice. Results are expressed as mean ± SD. * *p <* 0.05, ***p <* 0.01, *** *p <* 0.001, and **** *p <* 0.0001.

**Figure 4 nutrients-15-01909-f004:**
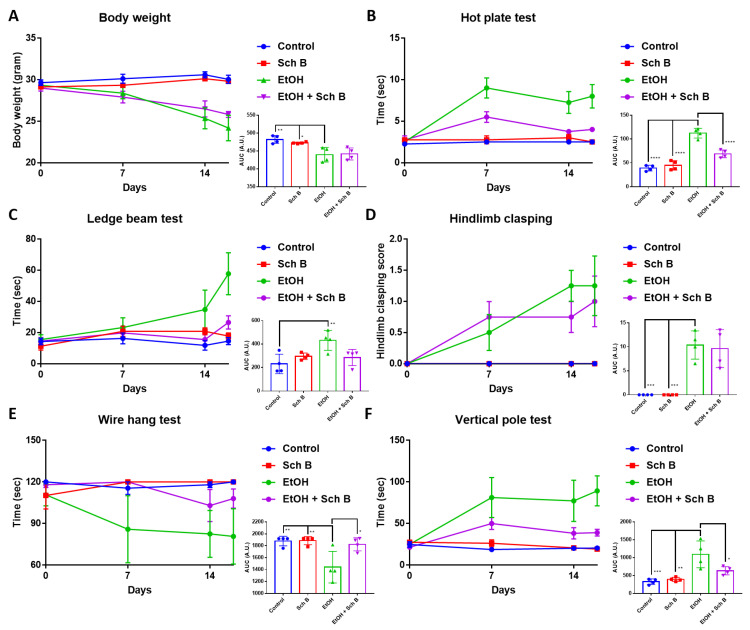
Schisandrin B prevents neurological function defects in mice consuming ethanol. (**A**) Mouse body weight. Assessment of neurological functions in mice by the (**B**) hot plate, (**C**) ledge beam, (**D**) hindlimb clasping, (**E**) wire hang, and (**F**) vertical pole tests. *n* = 4 mice. Results are expressed as mean ± standard error mean (SEM), and area under the curve (AUC) values are expressed as mean ± SD. * *p <* 0.05, ** *p <* 0.01, *** *p <* 0.001, and **** *p <* 0.0001.

**Figure 5 nutrients-15-01909-f005:**
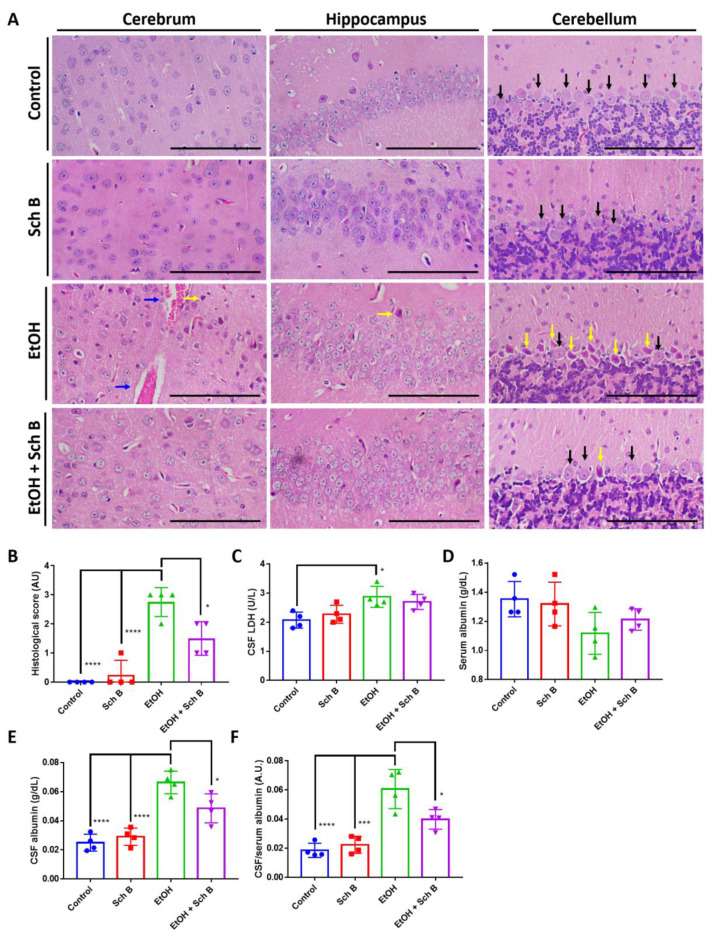
Schisandrin B prevents brain injuries in mice consuming ethanol. (**A**) Representative H&E-stained histological images showing the brain pathology of the mice. Congestion (blue arrows) and neuronal cell death (yellow arrows) were seen in ethanol-treated mice, which improved after Sch B treatment. Black arrows represent normal Purkinje fibers. Scale bar corresponds to 200 μm. (**B**) Histological scores based on the histological slides from (**A**). (**C**) LDH concentration in CSF. Levels of (**D**) serum albumin, (**E**) CSF albumin, and (**F**) CSF-to-serum albumin ratio. *n* = 4 mice. Results are expressed as mean ± SD. * *p <* 0.05, *** *p <* 0.001, and **** *p <* 0.0001.

**Figure 6 nutrients-15-01909-f006:**
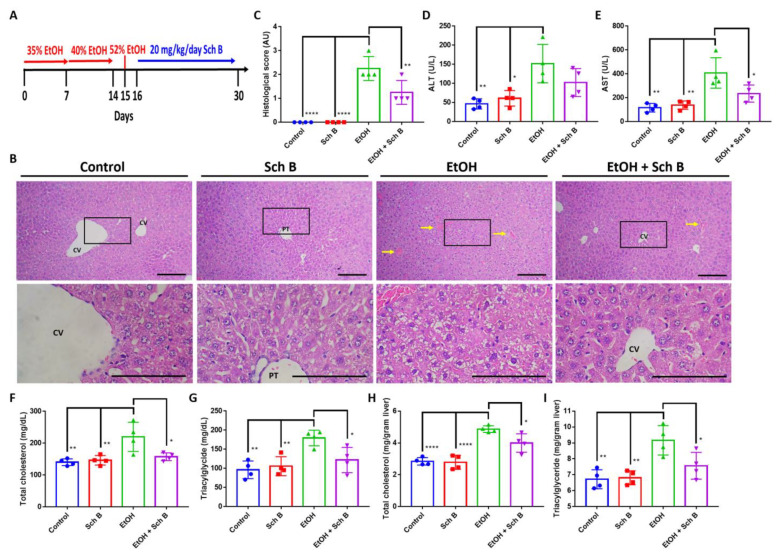
Schisandrin B alleviates ethanol-induced liver injuries in mice. (**A**) Experimental design scheme. (**B**) Representative H&E-stained histological images showing the liver pathology of the mice. Congestion (yellow arrows) and ballooning degeneration were seen in ethanol-treated mice, which were much improved by Sch B treatment. Scale bar corresponds to 200 μm. CV, central vein; PT, portal triad. (**C**) Histological scores based on the histological slides from (**B**). Serum levels of (**D**) ALT, (**E**) AST, (**F**) total cholesterol, and (**G**) triacylglycerides. Hepatic levels of (**H**) total cholesterol and (**I**) triacylglycerides. *n* = 4 mice. Results are expressed as mean ± SD. * *p <* 0.05, ** *p <* 0.01, and **** *p <* 0.0001.

**Figure 7 nutrients-15-01909-f007:**
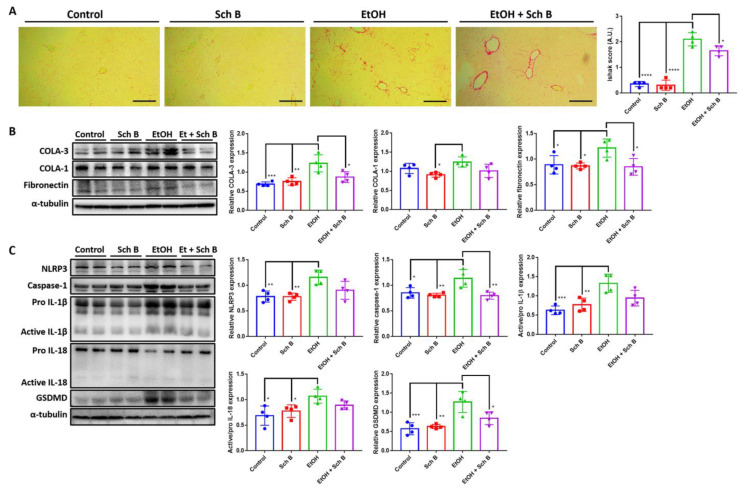
Schisandrin B alleviates ethanol-induced liver inflammasome activation and fibrosis. (**A**) Representative Sirius red-stained histological images showing collagen deposition (red color) in the mouse liver. Levels of fibrosis were quantified by Ishak score based on the Sirius red-stained slides. (**B**) Representative Western blot images and relative densitometric bar graphs of (**B**) fibrotic markers and (**C**) inflammasome components. Expression levels are standardized by α-tubulin or the pro-form protein levels as appropriate. *n* = 4 mice. Results are expressed as mean ± SD. * *p <* 0.05, ** *p <* 0.01, *** *p <* 0.001, and **** *p <* 0.0001.

**Figure 8 nutrients-15-01909-f008:**
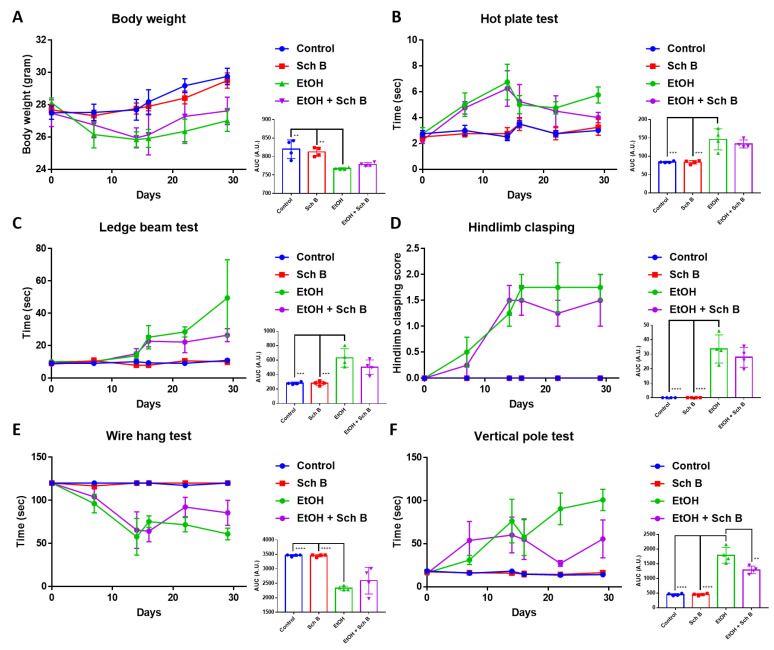
Schisandrin B repairs ethanol-induced neurofunctional defects. (**A**) Mouse body weight. Assessment of neurological functions in mice using the (**B**) hot plate, (**C**) ledge beam, (**D**) hindlimb clasping, (**E**) wire hang, and (**F**) vertical pole tests. *n* = 4 mice. Results are expressed as mean ± standard error mean (SEM), and area under the curve (AUC) values are expressed as mean ± SD. ** *p <* 0.01, *** *p <* 0.001, and **** *p <* 0.0001.

**Figure 9 nutrients-15-01909-f009:**
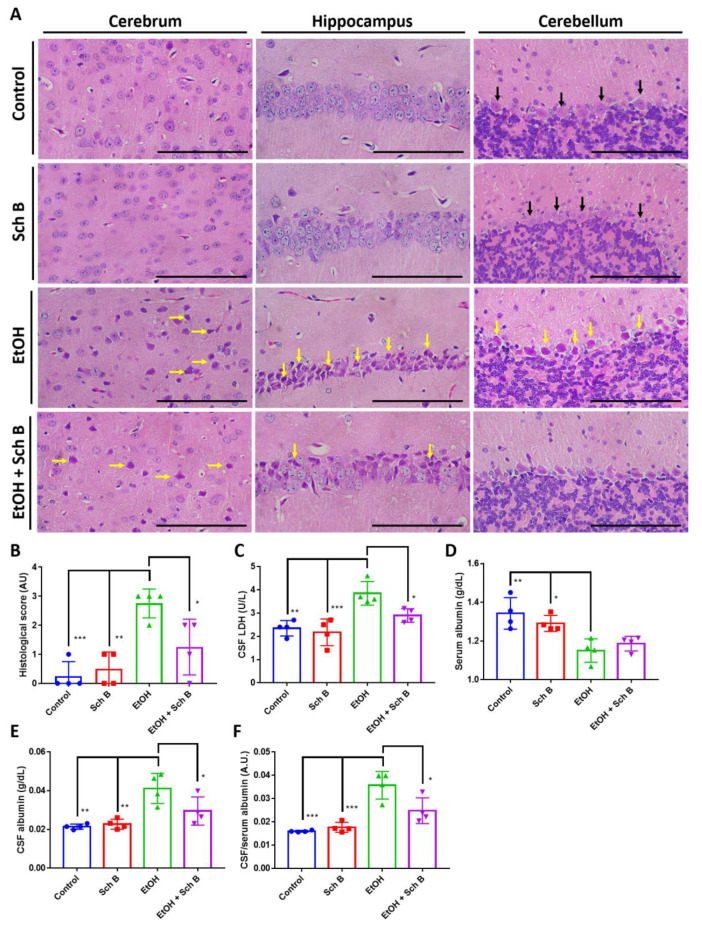
Schisandrin B alleviates ethanol-induced brain injuries in mice. (**A**) Representative H&E-stained histological images showing the brain pathology of mice. Neuronal cell death (yellow arrows) was seen in ethanol-treated mice, which improved after Sch B treatment. Black arrows represent normal Purkinje fibers. Scale bar corresponds to 200 μm. (**B**) Histological scores based on the histological slides from (**A**). Concentrations of (**C**) CSF LDH, (**D**) serum albumin, (**E**) CSF albumin, and (**F**) CSF-to-serum albumin ratio. *n* = 4 mice. Results are expressed as mean ± SD. * *p <* 0.05, ** *p <* 0.01, and *** *p <* 0.001.

## Data Availability

The datasets generated and analyzed during the current study are available from the corresponding authors upon reasonable request.

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
