# Peer review of "Prevention of the Pro-Aggressive Effects of Ethanol-Intoxicated Mice by Schisandrin B"

_nutrients, 2023, doi:10.3390/nu15081909_

Round 1
Reviewer 1 Report
The authors evaluated the effects of Schisandrin B (SchB) on preventing alcohol-induced liver and brain damage in two models, preventive and corrective. The authors determined various histological, biochemical, and molecular injury indicators in the liver and brain. SchB administration ameliorated liver damage, lipid deposition, inflammation, and fibrosis; also, SchB reversed brain damage provoked by ethanol intake, improving mice's neurological function.
There are some questions and suggestions for the authors:
a) Regarding both treatments, declare in the Materials and Methods section that the first is a preventive model and the second is a corrective one. Please, declare the complete name of all abbreviations used in the Western blotting subsection and throughout the manuscript, although these terms for proteins are well known. How were the doses of 20 and 25 mg/kg selected by preliminary studies or based on literature references? Provide the rationale for this.
b) The authors mention that mice underwent cardiac puncture to obtain the serum samples later; thus, the sacrifice occurs by exsanguination. However, was this procedure done under anesthesia or not? Please clarify this concern.
c) The chemical structure of SchB may be interesting to be added to one of the figures.
d) In reversion models of liver damage induced by oxidative stress as EtOH or carbon tetrachloride, there is a natural reversion once the toxic agent has been withdrawn; therefore, how was this determined in the reversion model? One can observe this natural reversion in a group that receives just a vehicle (water for EtOH) during the same period to compare this against the one administered with SchB. Were the four groups sacrificed after the 14 days or immediately sacrificed after receiving their particular treatments before the 14 days? Please clarify this doubt.
e) Given the powerful profibrotic effect of TGF-b, why was this not determined in the samples of serum or liver? Please offer the rationale for this. Besides, the Discussion section seems to be very short since no comparisons with other reports are made to contrast ALT/AST values as necrosis markers, the importance of each neurological and motor coordination test explaining similar results by administering SchB or others alike. Please go deeper into this issue.
Author Response
Response to Reviewer 1 Comments
The authors evaluated the effects of Schisandrin B (SchB) on preventing alcohol-induced liver and brain damage in two models, preventive and corrective. The authors determined various histological, biochemical, and molecular injury indicators in the liver and brain. SchB administration ameliorated liver damage, lipid deposition, inflammation, and fibrosis; also, SchB reversed brain damage provoked by ethanol intake, improving mice's neurological function.
There are some questions and suggestions for the authors:
- Regarding both treatments, declare in the Materials and Methods section that the first is a preventive model and the second is a corrective one. Please, declare the complete name of all abbreviations used in the Western blotting subsection and throughout the manuscript, although these terms for proteins are well known. How were the doses of 20 and 25 mg/kg selected by preliminary studies or based on literature references? Provide the rationale for this.
Response: Thank you for raising these important questions. We agree with you and have edited the manuscript.
Response: The two treatment models have been declared in the Materials and Methods section as well as throughout the manuscript. Examples such as:
Line 67: “Two treatment models (preventive model and corrective model) were employed…”
Line 68: “In the first model (preventive model),…”
Response: All the abbreviations have been declared in the manuscript.
(Line 102-115 and Line 128-133.)
Response: Thank you for pointing out the mistakes. 25 mg/kg is a typo error. All the concentration used in this study was 20 mg/kg. This has been corrected:
Line 79: “…mice were treated with 20 mg/kg Sch B by oral gavage for 14 consecutive days.”
Response: Treatment concentrations of Sch B (20 mg/kg) were based on literature references. This information has also been added in the Materials and Methods section.
Line 82-84: “Treatment concentrations of Sch B (20 mg/kg) were based on literature references [28,29]. No mice died or exhibited Sch B-related adverse effects during the experiment.”
- The authors mention that mice underwent cardiac puncture to obtain the serum samples later; thus, the sacrifice occurs by exsanguination. However, was this procedure done under anesthesia or not? Please clarify this concern.
Response: The mice were sacrificed by exsanguination without anesthesia. This information has been added.
Line 80-81: “Mice were sacrificed by exsanguination without anesthesia one day after the last Sch B treatment (day 30) with 12-h fasting.”
- The chemical structure of SchB may be interesting to be added to one of the figures.
Response: Thank you for the suggestion. The chemical structure of Sch B has been added in Figure 1.
- In reversion models of liver damage induced by oxidative stress as EtOH or carbon tetrachloride, there is a natural reversion once the toxic agent has been withdrawn; therefore, how was this determined in the reversion model? One can observe this natural reversion in a group that receives just a vehicle (water for EtOH) during the same period to compare this against the one administered with SchB. Were the four groups sacrificed after the 14 days or immediately sacrificed after receiving their particular treatments before the 14 days? Please clarify this doubt.
Response: Thank you for raising this important point. We confirmed that there was no natural reversion when ethanol treatments were removed. As in the second model (Figure 6), in the EtOH group, mice were fed with vehicle after ethanol treatment ended; whereas in the EtOH+Sch B group, mice were fed with Sch B after ethanol treatment ends. And the results suggested that there were still persistent liver injuries in the EtOH group (Figure 6), suggesting there was no natural reversion. This information was also added to the text.
Line 277-279: “Furthermore, by comparing the ethanol-fed group with the treatment group, we can observe that ethanol-induced injuries persisted, suggesting that there was no natural reversion once ethanol was removed.”
Response: The groups were sacrificed one day after the last treatment (on day 15). This information has been added.
Line 80-81: “Mice were sacrificed by exsanguination without anesthesia one day after the last Sch B treatment (day 30) with 12-h fasting.”
- Given the powerful profibrotic effect of TGF-b, why was this not determined in the samples of serum or liver? Please offer the rationale for this. Besides, the Discussion section seems to be very short since no comparisons with other reports are made to contrast ALT/AST values as necrosis markers, the importance of each neurological and motor coordination test explaining similar results by administering SchB or others alike. Please go deeper into this issue.
Response: Thank you for the suggestion. We have therefore performed an ELISA test targeting TGF-β levels in the serum and liver. The data have been added in Supplementary Figure 1. And related information has been added to the manuscript. (Line 133-135; 206-208; 293-295)
Response: Discussion regarding the liver functions and neurological functions has been added in the discussion part.
Line 353-379: “Several neurological tests were employed in this study: the hot plate test was used to measure thermal nociception in the mice; the ledged beam test is used to assess sensorimotor deficits of the mice; the hindlimb clasping score was used as an indicator of the severity of motor dysfunction; the wire hang test seeks to evaluate motor functions of the mice; and the pole test was employed to assess basal ganglia-related movement in mice [40]. Our results have suggested that Sch B treatment can beneficially prevent ethanol-induced neurological defects (Fig 4).In contrast to our study, in a mice model with Angiostrongylus cantonensis-induced meningoencephalitis, the sole use of Sch B fails to repair the mice neurological functions, but only if Sch B were used together with the standard parasitic treatment drug, Albendazole [40]. This could be because using Sch B alone can-not kill the parasites, therefore, even if Sch B ameliorates brain inflammation, neurological functions were not improved as injuries are persistent [40]. Another study using mice model with Schistosoma manosni (S. mansoni)-associated neurological deflects showed an-other therapeutic outcome by Sch B [41]. S. manosni is a parasitic worm that caused liver fibrosis, and liver fibrosis always leads to subsequence neurological damage [42]. In that study, the use of Sch B alone can already improve mice performances on neurological tests [41]. As Sch B resolves liver injuries and fibrosis in S. mansoni-infected mice, it led to lesser brain injuries, and thereby, improve the mice neurological functions. Along with neuro-logical functions, improvement of the brain pathology (Fig 5A-C) and BBB damages (Fig 5D-F) were also seen. Previously, Sch B has shown a beneficial role in several CNS diseases such as cerebral ischemia [43,44], cerebral artery occlusion [44], subarachnoid hemorrhage [45], chemical-induced brain injury [28], and parasite-induced brain injury [40]. These studies have suggested that Sch B may target inflammasome activation to ameliorate brain inflammation and injuries. Therefore, it is presumed that Sch B may not only hinder inflammasome activation to prevent ethanol-induced neurological injuries, but this preventive effect may also come as a result of the resolution of liver damages.”
Line 382-383: “This treatment effect on the liver has as well been observed in chemical-induced [28] and infection-related liver injuries [19].”
Reviewer 2 Report
In this study, the authors investigated the preventative and therapeutic effects of Schisandrin B (Sch B) against ethanol-induced liver and brain injuries. By using two treatment models, the authors showed that Sch B can effectively prevent and ameliorate alcoholic liver diseases such as resolving liver injuries, lipid deposition, inflammasome activation, and fibrosis. Sch B reverses brain damage and improves the neurological function of ethanol-treated mice. The authors demonstrated that Sch B may serve as a potential treatment option for liver diseases, as well as subsequential brain injuries. Overall, the findings provided by the authors could have interesting implications. Specific concerns are listed as following points:
1. In Figure 1 and 5, it’s necessary to evaluate the steatosis levels by staining, such as semi-quantified Oil Red O staining. H&E-stain is not sufficient to evaluate steatosis.
2. The albumin levels in CSF are comparably much lower than serum albumin levels. It would be great if the authors could discuss this point and mention if previous findings have demonstrated this difference.
3. The authors need to better explain the western blot data of GSDMD (pyroptosis) in the result section as this programmed cell death can be initiated by NLRP3.
4. The oral ethanol gavage given to the 2 treatment models are quite high concentrated. It would be necessary to clarify if there are any mice died during the treatments.
Author Response
Response to Reviewer 2 Comments
In this study, the authors investigated the preventative and therapeutic effects of Schisandrin B (Sch B) against ethanol-induced liver and brain injuries. By using two treatment models, the authors showed that Sch B can effectively prevent and ameliorate alcoholic liver diseases such as resolving liver injuries, lipid deposition, inflammasome activation, and fibrosis. Sch B reverses brain damage and improves the neurological function of ethanol-treated mice. The authors demonstrated that Sch B may serve as a potential treatment option for liver diseases, as well as subsequential brain injuries. Overall, the findings provided by the authors could have interesting implications. Specific concerns are listed as following points:
- In Figure 1 and 5, it’s necessary to evaluate the steatosis levels by staining, such as semi-quantified Oil Red O staining. H&E-stain is not sufficient to evaluate steatosis.
Response: Thank you for the reviewer’s suggestion. However, lipid staining may not suitable for paraffin-embedded tissue sections (which is what we used in this study). Unfortunately, performing this lipid staining at this point would warrant the additional euthanization of 32 new animals, which we believe will against the concept of the 3Rs.
Although we cannot evaluate steatosis by lipid staining, our current data such as cholesterol and TG levels in the liver, as well as in the serum may suggest lipid deposition in the liver. In addition, the method for ethanol treatment used in this study was employed as described by other literature, which has been shown to cause lipid deposition in the liver. Related references were added in the material and methods section (Line 81-82).
- The albumin levels in CSF are comparably much lower than serum albumin levels. It would be great if the authors could discuss this point and mention if previous findings have demonstrated this difference.
Response: Thank you for raising this point. Because CNS cannot synthesize albumin, albumin found in the CSF was derived from the blood (by crossing the BBB); therefore, albumin levels in the CSF are much lower than that in the serum. This difference has also been observed in other studies. We have added this information to the manuscript:
Line 247-249: “While albumin synthesis occurs in the liver but not in the CNS, albumin present in CSF usually derives from plasma [28,34]. The ratio of CSF and serum albumin was used to evaluate the blood-brain barrier (BBB) damage.”
- The authors need to better explain the western blot data of GSDMD (pyroptosis) in the result section as this programmed cell death can be initiated by NLRP3.
Response: Thank you for the suggestion. A description regarding GSDMD has been added to the manuscript.
Line 202-204: “GSDMD, a pyroptotic marker that can be initiated by NLRP3 signaling [33], has also been significantly increased in expression by ethanol, suggesting the presence of pyroptosis (Fig 3C).”
Line 344-346: “In addition, it initiates a form of programmed cell death called pyroptosis, which is driven by a protein called GSDMD [36].”
- The oral ethanol gavage given to the 2 treatment models are quite high concentrated. It would be necessary to clarify if there are any mice died during the treatments.
Response: Thank you for raising this point. The method for ethanol treatment used in this study was employed as described by other literature. Related references were added in the material and methods section (Line 81-82). And we have also added the information that no mice died during the experiment.
Line 81-84: “Method for ethanol treatment was employed as described previously [25-27]. Treatment concentrations of Sch B (20 mg/kg) were based on literature references [28,29]. No mice died or exhibited Sch B-related adverse effects during the experiment.”